# Mechanically interlocked networks cross-linked by a molecular necklace

Zhaoming Zhang[1], Jun Zhao[1], Zhewen Guo[1], Hao Zhang[1], Hui Pan[1], Qian Wu[1], Wei You[1], Wei Yu[1] & Xuzhou Yan [1]✉

Molecular necklaces have attracted much research attention due to their unique topological structures. Although numerous molecular necklaces with exquisite structures have been constructed, it remains a major challenge to exploit the functions and applications associated with their fascinating architectural and dynamic characteristics. Herein, we report a class of mechanically interlocked networks (MINs) cross-linked by a molecular necklace, in which multiple crown ethers are threaded on a hexagonal metallacyclic framework to furnish a cross-linker with delicate interlocked structures. The molecular necklace cross-linker possesses multiple peculiar advantages: multivalent interactions and rigid metallacycle framework guarantee robust features of MINs while the motion and dissociation of the interlocked structures bring in notable mechanical adaptivity. Moreover, the MINs could respond to the stimuli of $K^+$ and $Br^-$, which lead to the dethreading of crown ether and even the complete decomposition of molecular necklace, respectively, showing abundant active properties. These findings demonstrate the untapped potential of molecular necklaces as cross-linkers and open the door to extend their advanced applications in intelligent supramolecular materials.

[1] School of Chemistry and Chemical Engineering, Frontiers Science Center for Transformative Molecules, Shanghai Jiao Tong University, Shanghai 200240, P. R. China. ✉email: xzyan@sjtu.edu.cn

The implementation of many sophisticated functions in living organisms relies on their highly complex but organized architectures which are oftentimes formed via dynamic cross-linking of corresponding constituents. For example, the sliding motion of the thin filament during muscle contraction depends on a strong polymeric network wherein the thick filament serves as a cross-linker precisely and reversibly forms multiple cross bridges with thin filaments[1–4]. The typical features of such biomacromolecular cross-linkers could be summarized as delicate structures, multivalent interactions, and dynamic characteristics. Inspired by nature, it can be expected that synthetic cross-linkers possessing these merits would also be able to endow materials with sophisticated functions and emergent properties. Although numerous novel cross-linkers have been developed in the past decades[5–13], the construction of polymeric networks in which the cross-linkers, simultaneously have the advantageous features of delicate structures, multivalent interactions, and dynamic characteristics, is still an enormous challenge.

Molecular necklaces, as a special member of mechanically interlocked molecules[14–20], are derived from catenanes, in which a large macrocycle is threaded by a number of small rings (at least three). A molecular necklace consisting of $(n-1)$ small rings threaded on a large ring (total $n$ rings) is denoted as $[n]$MN[21]. The first molecular necklace was discovered by Sauvage et al. during the synthesis of [3]catenane[22]. Since then, molecular necklaces with diverse structures have been constructed[23–33]. For example, Kim and coworkers prepared a [6]MN based on intermolecular charge-transfer interactions stabilized by cucurbit[8] uril hosts (Fig. 1a)[34]. Stoddart et al. reported a densely charged [5]MN with four 4+ charged blue boxes mechanically interlocked around an 8+ charged large ring (Fig. 1b)[35]. Harada and coworkers synthesized a cyclodextrin-based [n]MN through a photodimerization of 9-anthracene-capped polyrotaxane (Fig. 1c)[36]. Recently, Yang et al. presented a heterometallic triangular necklace containing both Cu and Pt metals, which showed strong antibacterial activity (Fig. 1d)[37].

These exquisite architectures indicate that the structural complexity and synthetic chemistry for molecular necklaces have reached a high level. However, the exploitation of functions and applications associated with their unique structural characteristics is severely lagging behind their frameworks themselves, which represents an urgent issue for promoting the further development of molecular necklaces. From the point of view of topological structures, we envision that the molecular necklaces hold great potential as a bioinspired cross-linker for the construction of mechanically interlocked networks (MINs)[38–40] in terms of the following considerations: (i) Molecular necklaces bearing multiple small rings would enable multivalent cross-linking to form robust MINs. (ii) Multiple interlocked structures along with inherent noncovalent interactions in molecular necklaces would endow the MINs with abundant dynamic properties towards external stimuli, such as force and addition of chemicals. (iii) Molecular necklaces, especially those with bulky and rigid frameworks, could act as reinforcing components to strengthen the MINs. (iv) The delicate and complicated structures of molecular necklaces would give rise to cross-linked networks with novel topologies, thereby enriching the library of MINs. Nevertheless, mechanically interlocked networks cross-linked by molecular necklaces have yet to be achieved.

Herein, we report a de novo chemical design of MINs cross-linked by a molecular necklace in which the multivalent interlocked structures and the hierarchical organization of dynamic interactions endow the resultant MINs with mechanically robust yet stimuli-responsive properties. In specific, the molecular necklace consists of multiple dibenzo-24-crown-8 motifs (DB24C8, small ring) and a hexagonal metallacycle (large ring) formed by coordination-driven self-assembly (Fig. 2a, b), wherein the host−guest recognition and metal-coordination are orthogonal. As such, by mixing the independently formed metallacycle with DB24C8-decorated covalent polymer, the DB24C8 moiety threaded onto the metallacycle spontaneously to form the molecular necklace cross-linked MINs (Fig. 2c). Benefiting from the peculiar structural and dynamic characteristics, the molecular necklace plays versatile roles in the mechanical behaviors of the MINs such as multivalent interactions, the function as a rigid skeleton, as well as unique energy dissipation behaviors via force-induced mechanical motion and breaking of the interlocked structures. As a result, the representative MIN-**2** showed notable mechanical performance including stiffness (Young's modulus = 47.3 MPa), strength (breaking stress = 12.1 MPa), and toughness (49.3 MJ/m³). Moreover, we took advantage of K⁺-responsiveness of the host–guest recognition and Br⁻-responsiveness of the metal-coordination to modulate the mechanical properties of the MINs via dethreading the DB24C8 rings from the metallacycle and decomposition of the metallacycle, respectively.

## Results

**Design, synthesis, and structural characterization.** The polymer adopted in this work is a DB24C8-functionalized polynorbornene derivative prepared by ring opening metathesis polymerization (ROMP)[41,42] (Fig. 2a). The molecular necklace cross-linker in the MINs is constituted by couples of DB24C8 subrings threaded on a self-assembled metallacycle based on metal-coordination between di-platinum acceptor and bis(pyridinium) ligand (Fig. 2b, c), which was evolved from the discrete molecular necklace structure reported by Stang and cowokers[31]. By taking advantage of the orthogonal self-assembly between host–guest recognition and metal-coordination[43–45], the MINs were prepared based on the spontaneous formation of a molecular necklace. As depicted in Fig. 2c, the polymer with dangling DB24C8 moieties and the metallacycle were synthesized independently. When they were mixed in acetone, the reversible exchange of the metal-ligand bond allowed the DB24C8 moieties of the polymer threaded on the metallacycle, leading to the generation of MINs cross-linked by the spontaneously formed

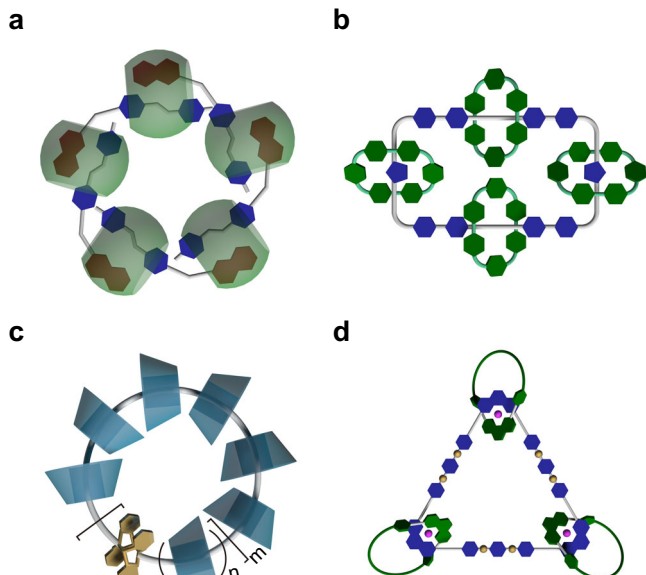

**Fig. 1 Cartoon representations of the selected previously reported molecular necklaces. a** [6]MN reported by Kim et al.[34], **b** [5]MN reported by Stoddart et al.[35], **c** [n]MN reported by Harada et al.[36], and **d** [4]MN reported by Yang et al.[37].

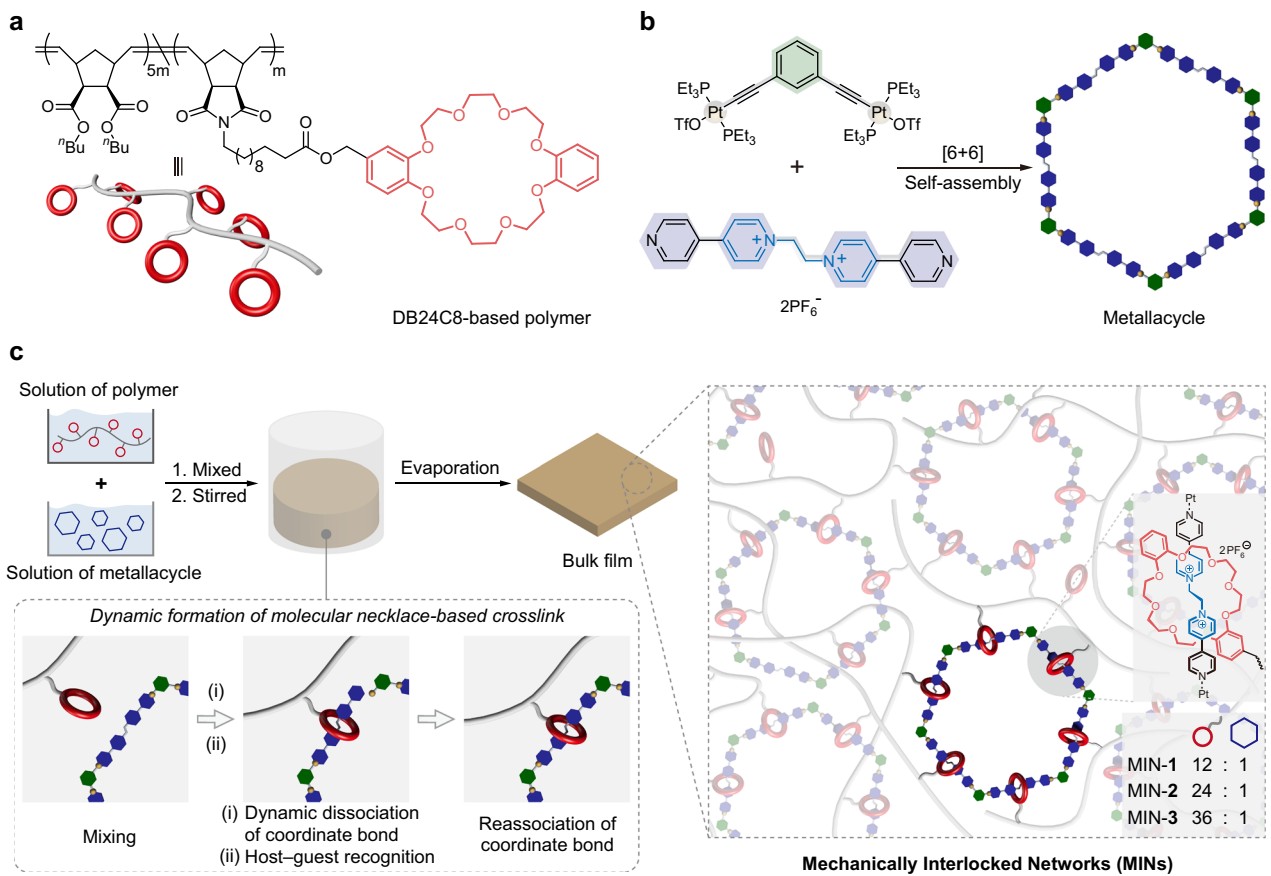

**Fig. 2 Preparation of the MINs. a** Chemical structure and corresponding cartoon of DB24C8-based polymer. **b** Schematic representation of the formation of the metallacycle via coordination-driven self-assembly. **c** Schematic representation of the preparation of the molecular necklace cross-linked MINs via dynamic host−guest recognition between DB24C8 moieties and metallacycle.

molecular necklace. After evaporation of the solvent, bulk materials based on MINs were obtained. Besides, the cross-linking density of the MINs could be tailored by controlling the content of the metallacycle. For this purpose, three MINs named MINs-**1**−**3** were prepared with the feed molar ratios of 12/1, 24/1, and 36/1 (DB24C8 moiety/metallacycle), respectively.

Firstly, $^1$H NMR spectra were measured to prove the formation of MINs. As shown in Supplementary Fig. 31b, the $H_1−H_4$ signals of the hexagonal metallacycle underwent significant shifts compared with those of the precursor of 1,2-bis(pyridinium) ethane (Supplementary Fig. 31a) and were similar to the reported results[31]. After being mixed with the polymer, a new set of peaks emerged due to the host–guest recognition (Supplementary Fig. 31c). Their chemical shifts were identical to those of the reported molecular necklace but the peak shapes were much broader, which can be ascribed to the embedding of the metallacycles in the polymer matrix. Based on these results, it can be concluded that the DB24C8 rings of the polymer were threaded on the metallacycle. Furthermore, measurements of diffusion-ordered NMR spectroscopy (DOSY) for the samples were also performed (Fig. 3a). The weight-average diffusion coefficient for the pure DB24C8-based polymer and MIN-**2** were estimated to be $3.75 \times 10^{-11}$ and $1.57 \times 10^{-11}$ m$^2$/s, respectively, which verifies the formation of molecular necklace cross-linked MIN because cross-linked network generally exhibited a smaller diffusion coefficient compared to its linear counterpart[46].

The above NMR results have revealed that the combination of metallacycle and DB4C8-based polymer successfully led to the formation of MINs in solution. Subsequently, we investigated

whether the network was retained in the solid-state. To this end, a swelling experiment for MIN-**2** was conducted (Fig. 3b), which showed a distinct swelling behavior in THF (swelling ratio = 4.8). As shown by the inset images of Fig. 3b, the size of the MIN-**2** specimen increased markedly after being soaked in THF for 30 min. These results confirmed that MIN-**2** had a prominent cross-linked structure in the solid-state. Afterward, small-angle X-ray scattering (SAXS) test was performed to disclose detailed information of the molecular necklace (Fig. 3c). The scattering curve was fitted well with the Beaucage model, which can be used to describe particles composed of sub-particles with two radii of gyration ($R_g$ for the large-scale particle and $R_s$ for the sub-particles)[47]. The model fit to the SAXS data gave $R_g$ and $R_s$ with the values of 7.01 and 1.02 nm, respectively, which were close to the diameters of metallacycle (6.70 nm) and DB24C8 (0.83 nm) (Supplementary Figs. 37, 38). Therefore, the SAXS results suggested that the molecular necklace was present in the bulk material.

Furthermore, the distribution of molecular necklaces in MINs was examined on a relatively large scale by EDS mapping. The elements such as carbon and oxygen for the polymer and phosphorus and platinum for the metallacycle were observed (Fig. 3d), indicative of the co-existence of the two components. Particularly, there was no distinct aggregation observed in the images of phosphorus and platinum elements, implying that the molecular necklaces were dispersedly anchored in the network.

**Mechanical properties of MINs and controls**. With the MINs in hand, then we will study their mechanical properties to verify the

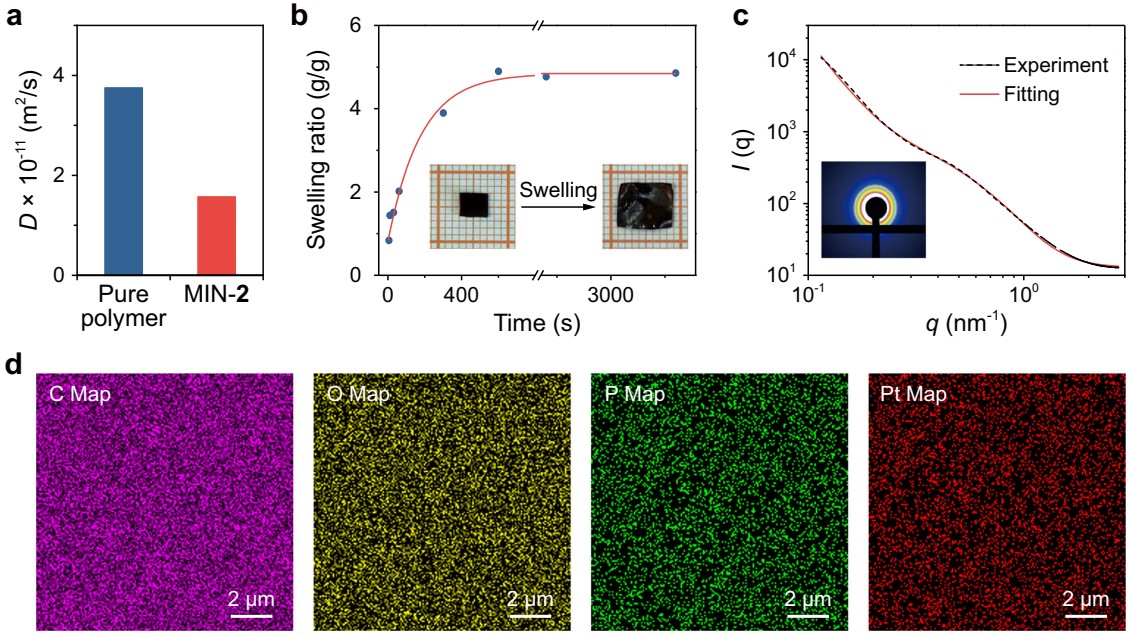

**Fig. 3 Characterization of the formation of molecular necklace-based cross-linker both in solution and in the solid-state. a** Diffusion coefficients of DB24C8-based polymer and MIN-**2** determined by DOSY NMR (400 MHz, acetone-$d_6$, 298 K). **b** Swelling ratio of the MIN-**2** in THF as a function of swelling time, and corresponding photographs (inset) before and after being soaked for 30 min. **c** SAXS curve for MIN-**2** and Beaucage model used to fit the SAXS data. **d** EDS mapping (carbon, oxygen, phosphorus, and platinum) for MIN-**2** neat film.

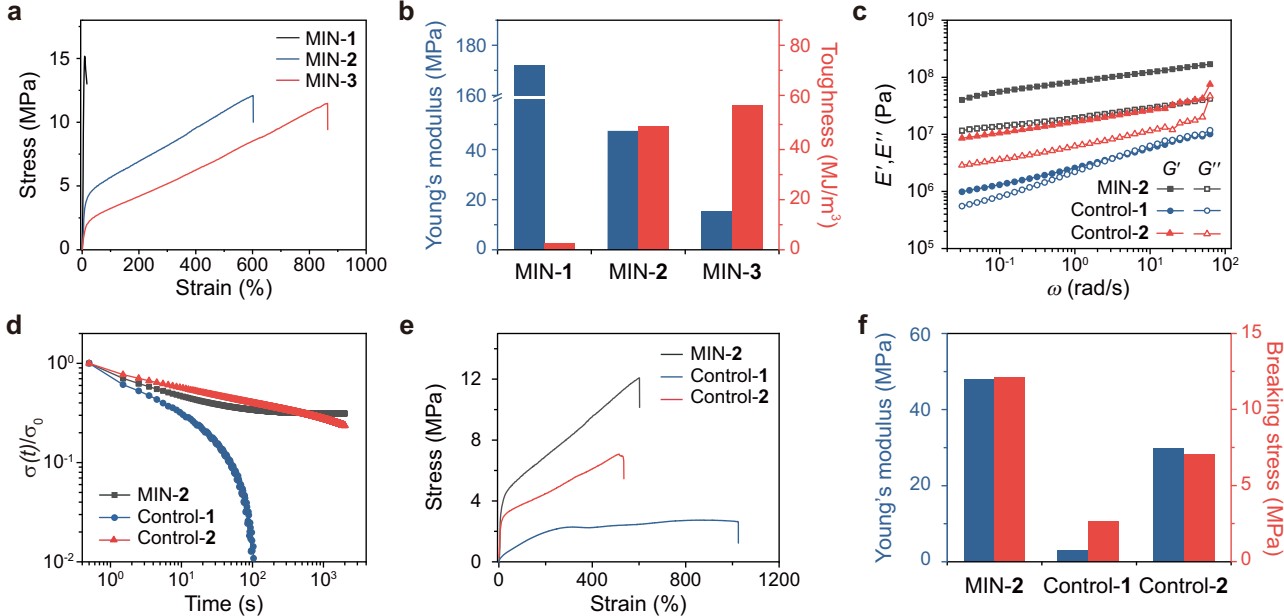

**Fig. 4 Mechanical properties of MINs and controls. a** Stress–strain curves of MINs-**1**−**3** were recorded with a deformation rate of 100 mm/min. **b** Young's moduli and toughness of MINs-**1**−**3** calculated from their stress–strain curves. **c** Frequency sweep of MIN-**2** and two controls measured by DMA. **d** Normalized stress relaxation curves of MIN-**2** and two controls measured at 30 °C. **e** Stress−strain curves of MIN-**2** and two controls recorded with a deformation rate of 100 mm/min. **f** Young's moduli and breaking stress of MIN-**2** and two controls calculated from their stress–strain curves.

performance of molecular necklaces as a cross-linker. Tensile tests of MINs-**1**−**3** showed that Young's moduli of the samples increased with the cross-linking density, and the values are calculated to be 172, 47.3, and 15.5 MPa for MIN-**1**, -**2**, and -**3**, respectively (Fig. 4a, b). The strain at the break had an opposite trend with the values of 9.20% for MIN-**1**, 604% for MIN-**2**, and 864% for MIN-**3**. Except for MIN-**1** with a brittle-hard property, both MIN-**2** and MIN-**3** exhibited decent toughness with values of 49.3 and 57.8 MJ/m³, respectively. These results demonstrated

that the density of molecular necklace-based cross-linking points played an important role in determining the mechanical properties of MINs. Overall, MIN-**2** had a moderate performance compared with the others and thus was selected for further study in the following sections.

To better showcase the peculiar cross-linking role of molecular necklace, two control samples were also designed. In the structure of control-**1** (Supplementary Fig. 3), its components are similar to MIN-**2** except that the platinum is in the form of

diiodo-precursor which cannot coordinate with bis(pyridinium) ligand (Supplementary Fig. 33), and thus there are no significant cross-linking in its structure. As for control-**2**, a complex composed of a di-platinum acceptor coordinated with two bis(pyridinium) ligand (namely, a fragment of the molecular necklace cross-linker) has been utilized to cross-link the DB24C8-functionalized polymer through the formation of [2]pseudorotaxane moiety (Supplementary Fig. 4), which could be regarded as a general supramolecular cross-link. Frequency sweep tests for MIN-**2** and two controls were firstly performed (Fig. 4c). The storage moduli ($E'$) of MIN-**2** and control-**2** were always larger than their loss moduli ($E''$) over the entire frequency range, showing high resistance to flow. On the contrary, the control-**1** had comparable elastic and viscous moduli in a broad frequency range with a distinct frequency dispersion. As mentioned above, the formation of a coordinate bond was not possible in the structure of control-**1**, and the frequency dispersion might stem from a slight entanglement of the polymer chains. In addition, the elastic moduli of the three samples showed the following order: MIN-**2** > control-**2** > control-**1**, indicating the high stiffness of MIN-**2**. Furthermore, stress relaxation was employed to reveal the structural features of the three samples. As shown in Fig. 4d, the stress applied on the control-**1** was released instantaneously and almost completely relaxed in about 100 s, whereas the stress relaxation for MIN-**2** and control-**2** were much slower, and finally, part of stress was kept. These results of frequency sweep and stress relaxation experiments further proved the effective cross-linking role of a molecular necklace in MIN-**2**, and also confirmed the structures of the two controls: control-**2** is a cross-linked network but no significant cross-linking exists in control-**1**.

More striking contrasts were shown by the stress−strain curves of the three samples (Fig. 4e). The Young's moduli were 47.3 MPa for MIN-**2**, 3.02 MPa for control-**1**, and 29.9 MPa for control-**2** (Fig. 4f), which were consistent with the tendency shown by the frequency sweep. The comparison between the two controls indicated the important role of cross-linking in enhancing mechanical properties. Meanwhile, although both control-**2** and MIN-**2** had cross-linked structures, MIN-**2** exhibited a much superior mechanical performance. Considering the fact that the evaluation of Young's modulus was based on the linear region wherein the molecular necklace cross-linker should keep intact, the enhanced stiffness of MIN-**2** might result from the multivalent interactions and the reinforcing effect of the rigid metallacycle[48]. Similar to the results of Young's modulus, the breaking stress of MIN-**2** (12.1 MPa) was also much higher than those of control-**1** (2.64 MPa) and control-**2** (7.09 MPa). The multivalent interactions and the reinforcing effect of the rigid skeleton may still be responsible for the superior strength of MIN-**2**, but the unique dynamicity of the molecular necklace made a contribution at the same time (vide infra). Besides, in terms of toughness, MIN-**2** had the best performance as well, which may be also strongly affected by the force-induced dynamicity of the molecular necklace as discussed in the following section.

### Force-induced dynamicity of molecular necklace in MINs.
Among the multiple roles of molecular necklace cross-linker, its dynamic behavior upon force is a crucial feature and has a significant effect on the mechanical properties of MINs. Stress−strain curves of MIN-**2** at different deformation rates were displayed in Fig. 5a. Pronounced deformation rate dependence of the mechanical behaviors was observed. Such a phenomenon was generally ascribed to the involvement of dynamic bonds in the system[49–54]. As for MIN-**2**, the dynamic bonds should be attributed to the supramolecular interactions of host–guest recognition and/or metal-coordination in molecular necklaces

(vide infra). Subsequently, the force-responsive behaviors of the two kinds of supramolecular interactions and even the whole molecular necklace are discussed below.

According to the literatures, the association constant of host–guest recognition is supposed to be much lower than that of the metal-coordination between platinum and pyridine ligands[55,56]. Such an estimation was also supported by the stress relaxation experiments at different temperatures for MIN-**2** and control-**2**. As shown in Fig. 5b, extraordinary thermal stability was shown by the MIN-**2** which remained notable applied force even at a temperature of 120 °C, and complete relaxation occurred until the temperature was up to 160 °C. By contrast, a rapid stress relaxation was observed for control-**2** at the temperature of 120 °C (Supplementary Fig. 43). The complete stress relaxation in these two samples should be relevant to the cleavage of the physical cross-links and consequent fracture of network structures. Hence, the dissociation of host–guest recognition in control-**2** would take place at the temperature of 120 °C. As for MIN-**2**, though host–guest recognition also dissociated at such high temperatures, the DB24C8 wheel was still locked by the intact metallacycle, thus maintaining an effective network. These results further verified the higher stability of metal-coordination compared with that of host–guest recognition, and thereby it is reasonable to speculate that upon applied force, the dissociation of host–guest recognition in the molecular necklace would occur firstly.

Then we wondered what would happen for the molecular necklace cross-link after the dissociation of host–guest interaction. To this end, recovery experiments for MIN-**2** were conducted (Fig. 5c). The initial circle exhibited a large hysteresis area, indicative of an efficient energy dissipation enabled by dynamic behaviors in a molecular necklace. In contrast, the second circle without rest had a significant residual strain as well as a markedly reduced hysteresis loop due to the delaying recovery of the dissociated supramolecular interactions. The IR spectrum of the sample after being stretched emerged a new peak around 669 cm$^{-1}$ which could be assigned as out-of-plane C-H bending vibrations of the $H_1$ on bis(pyridinium) ligand (Supplementary Fig. 45). This result indicated that dissociation of the coordination between bis(pyridinium) ligand and platinum took place during the stretch. However, the IR results also suggested that the dissociated coordinate bond could basically restore at 50 °C for 5 min. Although the reassociation of the coordinate bond was efficient, the recovery of mechanical properties didn't occur synchronously. For example, Young's modulus of the specimen only recovered 79% at 50 °C even the rest time was extended to 30 min (Supplementary Fig. 46). We deduced that the unrecovered mechanical properties should be more likely due to the failed recovery of partial host–guest recognition because the DB24C8 wheels pulled away from the metallacycle were difficult to rethread on the metallacycle in the solid-state. Therefore, after the dissociation of host–guest interaction, the DB24C8 moiety moved away from the binding site, which was also accompanied by the occurrence of the dissociation of a coordinate bond.

Furthermore, assisted by constrained geometries simulate external force (CoGEF) simulation[57], the unique dynamic features of the molecular necklace cross-link reflected by the above experiments could be deeply understood. The CoGEF elongation energy curve and corresponding optimized structures for a segment of the molecular necklace were shown in Fig. 5d and Supplementary Fig. 47, respectively. According to the elongation profile, the dynamic process of the molecular necklace could be roughly divided into three main stages. The initial stage (0–5 Å) with increased energy could be ascribed to the gradual dissociation of host–guest recognition, which is in accord with the

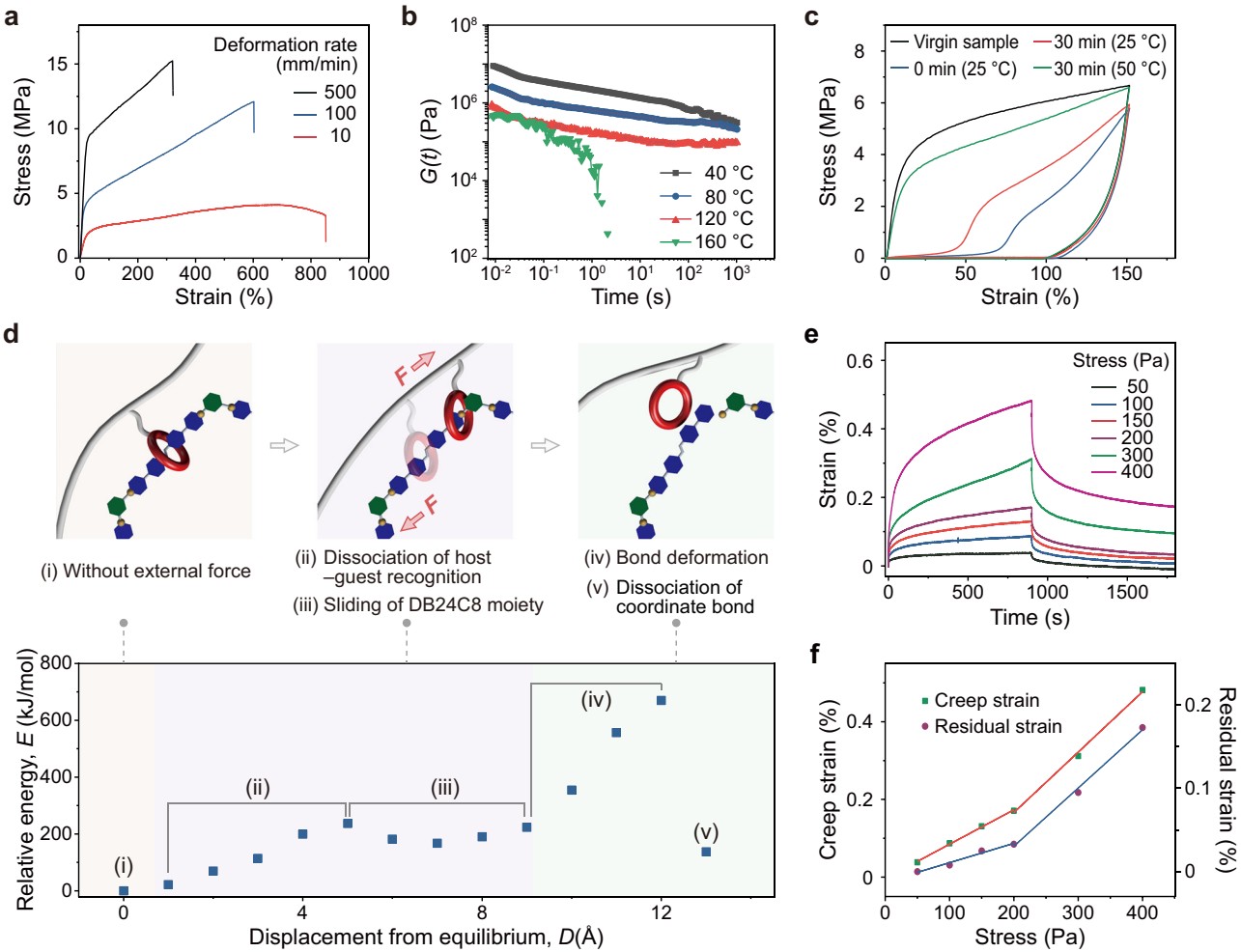

**Fig. 5 Force-induced dynamicity of molecular necklace in MINs. a** Tensile tests of MIN-**2** under different deformation rates ranging from 10 to 500 mm/min. **b** Temperature-dependent stress relaxation behaviors of MIN-**2**. **c** Cyclic tensile test curves of MIN-**2** loaded at a strain of 150% with different rest intervals and temperatures. **d** CoGEF potential as a function of stretched distance for a segment of a molecular necklace (bottom), and corresponding schematic representation of the force-induced dynamicity (top). **e** Creep-recovery curves of MIN-**2** with different stress levels at 30 °C. **f** Plots of creep and residual strains under different stress levels.

experimental fact that the dissociation of host–guest recognition occurs firstly upon stress. In the intermediate stage (5–9 Å), the elongation profile is relatively stable, and this stage corresponds to the sliding of the DB24C8 wheel on the bis(pyridinium) axle after the dissociation of host–guest recognition[58–61]. Subsequently, a sharp increase in energy was observed at the terminal region of the elongation curve (9–12 Å). In this stage, the wheel DB24C8 reaches the site of platinum which bears two triethylphosphine groups and is bulky enough to serve as a stopper. And this terminal stage is an enthalpic regime where bond deformation takes place. After the culmination of this regime, dissociation of metal-coordination occurs. Therefore, the dynamic process could be summarized as that the molecular necklace successively undergoes dissociation of host–guest recognition, sliding of DB24C8 wheel, and dissociation of metal-coordination in a synergistic fashion upon stress, which could efficiently equalize force and dissipate energy, and thereby imparts mechanical robustness and adaptivity to our MINs. In addition, the advantages of such a synergistic dynamicity were further highlighted through its comparison with control-**2**. The elongation curve for the cross-link in control-**2** was roughly equal to the first and second stages in that of molecular necklace, thus possessing shorter stretched distance and lower dissociation

energy (Supplementary Fig. 49). It means that the same energy which cleaves the cross-linking point in control-**2** is also able to dissociate the host–guest recognition in a molecular necklace, but the coordinate bond keeps intact. Under the circumstances, the DB24C8 moiety is still locked in the metallacycle with unchanged cross-linking density for MIN-**2**, and much higher energy is required to destroy the cross-linked network. As such, our MIN-**2** exhibited higher toughness and strength (Supplementary Fig. 50).

More information about the mechanical responsiveness of molecular necklace to external stress was provided by creep and recovery experiments under different applied loads (Fig. 5e). A typical character of the results was that: with the increase of loads, creep strains gradually increased and the residual strains became more and more noticeable upon stress-release at the same time. This tendency was well reflected in Fig. 5f, which summarized the creep and residual strains for all applied loads. Notably, both creep and recovery processes had a marked transition at the stress of 200 Pa, suggestive of the occurrence of a structure change. On the basis of the above analyses, this structure change may be relevant to the dynamic dissociation of a molecular necklace. Such a prominent transition further indicated the important role of the dynamicity of molecular necklace in the mechanical performance of MINs.

Combining the mechanical property study and the analyses of the force-induced dynamicity, the roles and advantages of the molecular necklace in cross-linking could be comprehensively revealed. Apart from the cross-linking role in a general sense, molecular necklace cross-linker enables to ensure a more robust network on the basis of its peculiar architectural and dynamic features: the multivalent cross-linking and reinforcing effect of rigid metallacycle endow the MINs with high stiffness, and the dynamicity of the multiple interlocked structure guarantees a notable toughness. Meanwhile, all these three features are responsible for the strength of MINs. For this reason, the stress–strain curves have exhibited that the mechanical properties of MIN-**2** are much superior to those of control-**2** in all aspects including stiffness, strength, stretchability, and toughness (Fig. 4e). These roles of molecular necklace cross-link have also been confirmed by further investigating the swollen MIN-**2** which is advantageous to weaken the effect of inter-chain interactions (Supplementary Fig. 42). As we mentioned above, control-**2** can be considered as a typical supramolecular cross-linked network. Thus, the comparison indicates that under equal conditions, the molecular necklace cross-linker could enhance the mechanical properties more efficiently compared with general supramolecular cross-linking. Moreover, supramolecular networks often suffer from poor stability due to the nature of weak noncovalent interactions. Reflecting on the stress relaxation experiments (Fig. 5b), the molecular necklace cross-linked MINs could overcome this weakness to a certain extent by showing good stability. Therefore, compared with general supramolecular cross-linking points, molecular necklace cross-link possesses multiple unique structural and dynamic characteristics to result in a more robust network, representing a more advanced dynamic cross-linking.

**Stimuli-responsive properties of MIN-2.** In the preceding sections, we have demonstrated the delicate architecture of the molecular necklace as a cross-linker facilitates the formation of MINs with favorable mechanical performance. To realize the imitation of bio-cross-linkers, stimuli-responsiveness of the molecular necklace cross-linkers is an important indicator. In the section of synthesis, the MINs were formed spontaneously by simply mixing the polymer and the hexagonal metallacycle in solution, which represented a kind of dynamic performance. As such, we wondered whether the DB24C8 moieties are able to dissociate dynamically from the interlocked structures of molecular necklaces via external stimuli. To verify this, MIN-**2** was treated with $KPF_6$ whose $K^+$ could compete with the bis(pyridinium) ligand to be recognized by the DB24C8 moieties. The $^1H$ NMR spectra showed that the signals assigned as the complexed peaks of bis(pyridinium) ligand and DB24C8 completely disappeared after being treated by $KPF_6$, and the bis(pyridinium) ligand signals were roughly similar to those of the pure metallacycle (Supplementary Fig. 51). Hence, the DB24C8 moieties were successfully separated from the molecular necklace but the metallacycle was still kept intact in the polymer matrix (Fig. 6d, left). Corresponding stress–strain curve of the treated sample showed that its mechanical properties had a sharp decrease compared with those of the virgin sample on account of the decross-linking of the molecular necklace (Fig. 6a). For example, Young's modulus declined from 47.3 to 26.2 MPa, and the toughness reduced from 49.3 to 21.9 $MJ/m^3$ (Fig. 6c). It is worth noting that the $K^+$-treated sample showed higher Young's modulus and breaking stress than those of the control-**1**, which might be caused by the skeleton effect of the rigid metallacycle worked as a reinforcing component (Supplementary Figs. 52, 53).

Apart from the host–guest recognition, the labile Pt−N bond in the molecular necklace is also easy to be destroyed by the

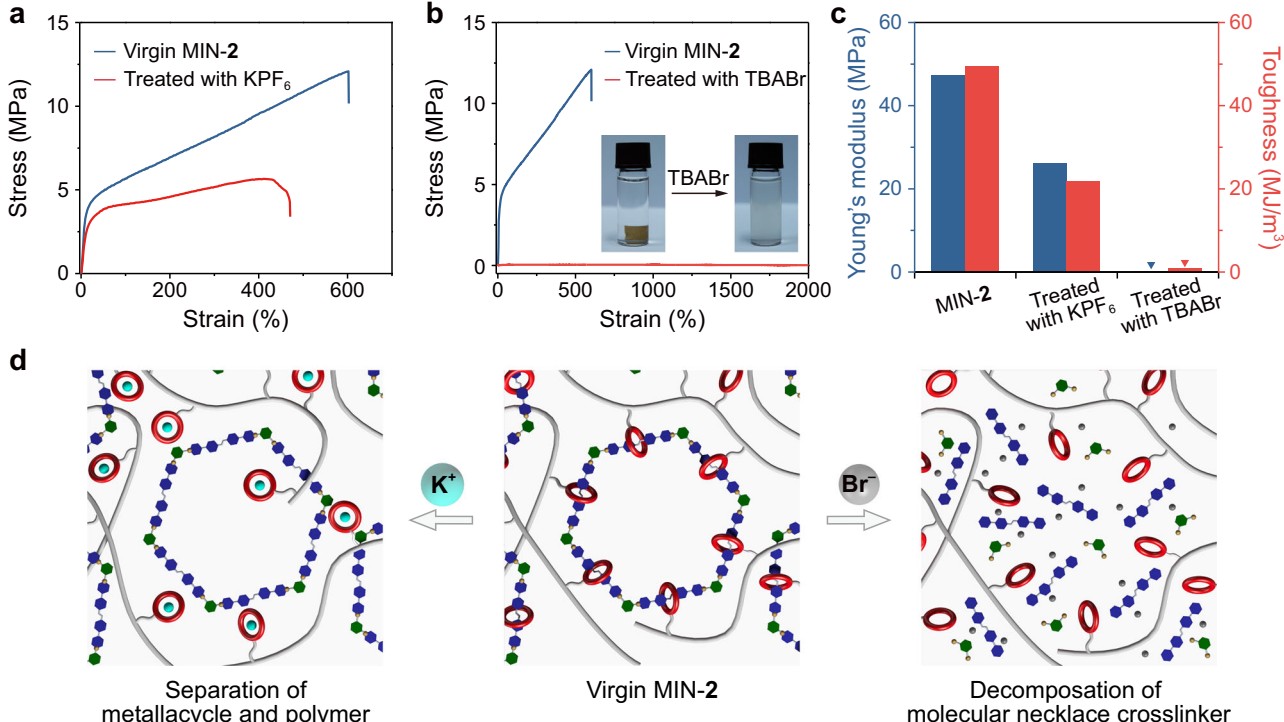

**Fig. 6 Stimuli-responsive properties of MIN-2. a** Stress–strain curves of the virgin MIN-**2** and the one after being treated with $KPF_6$. **b** Stress–strain curves of the virgin MIN-**2** and the one after being treated with TBABr. Inset: dissolution of MIN-**2** in acetone in the presence of TBABr. **c** Young's moduli and toughness of virgin MIN-**2** as well as the $KPF_6$ and TBABr-treated samples calculated from their stress–strain curves. **d** Schematic representation of the separation of DB24C8-based polymer and metallacycle triggered by $KPF_6$ (left), and the decomposition of molecular necklace triggered by TBABr (right).

introduction of competitive bromide[62]. For example, after being treated with tetrabutylammonium bromide (TBABr), the sample had very poor mechanical performance in light of stiffness (0.21 MPa), strength (0.05 MPa), and toughness (0.85 MJ/m$^3$), exhibiting a sharp contrast with the virgin one (Fig. 6b, c). The dramatic decrease of the mechanical properties could be explained that the addition of TBABr not only destroyed the coordinate bond but also dissociated the host–guest recognition owing to the exchange of counterion from PF$_6^-$ to Br$^-$, and thus led to the complete decomposition of the molecular necklace (Fig. 6d, right). Furthermore, the specimen of MIN-**2** could be dissolved in acetone in the presence of TBABr rather than swelled (Fig. 6b, inset), which was in accord with the decomposition of the molecular necklace.

## Discussion

In this work, we have designed and constructed a class of mechanically interlocked networks cross-linked by a molecular necklace in which multiple DB24C8 moieties are threaded on the metallacyclic core. Due to the unique dynamic interactions of the molecular necklace, the MINs are prepared in a convenient method by simply mixing hexagonal metallacycle and polymer decorated with DB24C8 moieties. Molecular necklace as a cross-linker integrates multiple advantages into the MINs to improve their mechanical properties. Multivalent interactions based on the multiple interlocked structures and the role of skeleton originated from the rigid and bulky metallacycle guarantee the stiffness and strength of the network. Furthermore, the force-induced motion of the interlocked structures and dissociation of the dual supramolecular interactions provide a unique and effective energy dissipation pathway to further reinforce the robustness of MINs. As a result, the MINs exhibit good mechanical performance in terms of stiffness (Young's modulus = 47.3 MPa), strength (breaking stress = 12.1 MPa), and toughness (49.3 MJ/m$^3$). In addition, it is disclosed that the hexagonal metallacycle could be separated from the molecular necklace by the introduction of competitive K$^+$, and also be completely decomposed in the presence of Br$^-$, exhibiting abundant dynamic and responsive properties. We believe that the molecular necklace cross-linked MINs hold great promise for the development of mechanically robust yet adaptive supramolecular materials, particularly, for those with precise structures and functions.

## Data availability

The authors declare that the data supporting the findings of this study are available within the paper and its supplementary information files or the data are available from the corresponding authors on reasonable request. Source data are provided with this paper.

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

## Acknowledgements

X.Y. acknowledges the financial support of the NSFC/China (22122105, 22071152, and 21901161) and the Natural Science Foundation of Shanghai (20ZR1429200). Z.Z. acknowledges the financial support of the NSFC/China (22101175). W.Yu acknowledges the financial support of the NSFC/China (51625303). This research is supported by the Starry Night Science Fund of Zhejiang University Shanghai Institute for Advanced Study, Grant No. SN-ZJU-SIAS-006.

## Author contributions

X.Y. supervised this research; X.Y. and Z.Z. conceived the project. Z.Z. carried out the synthesis and some characterization of the materials. Q.W. synthesized some precursor molecules. Z.G. and H.P. performed SAXS measurements and the associated analysis. Z.G. performed the CoGEF calculations. H.Z. and W.You conducted a rheological test under the supervision of W.Yu. The manuscript was written by Z.Z., J.Z., W.Yu, and X.Y. with contributions from all the coauthors.

## Competing interests

The authors declare no competing interests.
