## [Peer Review File · Nature Communications]

Mechanically interlocked networks cross-linked by a molecular necklaceREVIEWER COMMENTS

Reviewer #1 (Remarks to the Author):

This is a study of the mechanical and stimuli-response properties of elastomers fabricated using molecular necklaces as cross-linking points. The fact that the molecular necklace is a crosslinking point is extremely interesting. The mechanical properties suggest that the molecular necklaces are the crosslinking points, since the Young's modulus and breaking stress increase and the elongation at break decreases with the increase in the number of crosslinking points (increase in the number of molecular necklaces). At MIN-3, where toughness increases, sacrificial rupture of crosslinking points is suggested, and Fig. 5 shows a dynamic system of crosslinking points. However, these are not new features of materials with molecular necklaces as crosslinking points, as similar results can be obtained with elastomer and gel systems obtained from conventional dynamic bonding. This is also true for stimuli-responsive properties shown in Fig. 6. Comparing the mechanical strength in Fig. 6, it is clear that not necessarily only the crosslinking points by molecular necklaces affect the mechanical strength of the elastomer. It is important to clarify how efficiently the molecular necklaces are formed and to what extent they affect the mechanical properties.

Minor comments

Evaluation of mechanical strength of organogel swollen with THF helps to understand the correlation between structure and mechanical properties.

The correlation between sample name and curve in DSC in Fig.S28 is not reasonable.

Reviewer #2 (Remarks to the Author):

This manuscript reports a mechanically interlocked networks (MIN) by crosslinking a linear polymer with crown ether side chains and metallacycle via coordination-driven self-assembly. Dynamic crosslinks are formed by host-guest interaction and dynamic dissociation and reassociation of macrocyclic coordination bonds. The mechanically interlocked network shows various peculiar mechanical responses based on the formation, sliding motion, and dissociation of dynamic crosslinks. Since the manuscript is well written with the unique mechanically interlocked structure, I think that it might be published in Nature Communication. However, it seems like there are some unclear explanations about the mechanical data.

1) The authors said that no significant cross-links exist in the control sample. Certainly, the stress relaxation in Fig. 4d indicates that but the viscoelastic profile of the control sample in Fig. 4c shows a frequency dispersion like physical crosslinks, which is different from that of the polymer melt, in the frequency range measured. The same frequency dependence is observed in MIN-2 as well. The authors should explain the origin of the frequency dispersion.

2) In the molecular mechanism of dynamic crosslinks in MIN in Fig. 5e, I feel that there are no clear evidence supporting the second process: the dissociation of host-guest recognition and sliding. From the mechanical data, it seems to be difficult to discriminate between the second and third processes. Is it possible to explain which data clearly shows the existence of the second process?

Point-by-Point Response to Reviewers' Comments

For Reviewer 1:

1. This is a study of the mechanical and stimuli-response properties of elastomers fabricated using molecular necklaces as cross-linking points. The fact that the molecular necklace is a crosslinking point is extremely interesting. The mechanical properties suggest that the molecular necklaces are the crosslinking points, since the Young's modulus and breaking stress increase and the elongation at break decreases with the increase in the number of crosslinking points (increase in the number of molecular necklaces). At MIN-3, where toughness increases, sacrificial rupture of crosslinking points is suggested, and Fig. 5 shows a dynamic system of crosslinking points. However, these are not new features of materials with molecular necklaces as crosslinking points, as similar results can be obtained with elastomer and gel systems obtained from conventional dynamic bonding. This is also true for stimuli-responsive properties shown in Fig. 6.

- We would like to thank the reviewer for these valuable comments. Fundamentally, our work is aimed at utilizing the structural and dynamic characteristics of molecular necklace to crosslink polymers and developing a fully new topological network. Hence, the core objective in our work is to construct MINs cross-linked by molecular necklace and prove the applicability of molecular necklace as a cross-linker, especially, verify the important roles of the structural and dynamic characteristics of molecular necklace in determining the properties of the MINs. The results in our work demonstrate that all these goals have been achieved. Thus, we report the first MINs cross-linked by a molecular necklace, and also realize the first application of molecular necklace associated with its peculiar structural and dynamic characteristics in the field of elastomeric materials since the first molecular necklace was prepared 30 years ago (Sauvage, J. P. *et al. J. Am. Chem. Soc.* **1991**, *113*, 4023–4025). These achievements represent a great step forward in the development of molecular necklaces and mechanically interlocked polymers.

The reviewer mentioned that enhanced mechanical properties and stimuli-responsive properties could also be obtained by elastomers or gels with conventional dynamic bonding. It is true that these macroscopic properties as basic characters are easily to be simultaneously achieved by general supramolecular cross-linkers. Moreover, compared with other features, these properties are also the most important indicators when evaluate the effect of dynamic cross-links. Therefore, for the development of new dynamically cross-linked systems, better mechanical properties and dynamic performances are always pursued. Given this perspective, the emergence of molecular necklace cross-linked MINs is very meaningful to the development of dynamic networks. Firstly, molecular necklace cross-linked MINs represent a totally new dynamic network with unique topological structure, which provides a new idea in developing dynamically cross-linked polymers. What's more, compared with general supramolecular cross-linking points, the molecular necklace as a cross-link has its unparalleled merits in the aspects of mechanical properties and responsive behaviors, which are summarized as follows:

- (1) Under equal conditions, molecular necklace cross-linker could endow the polymeric system with better mechanical performance and higher stability compared with

general supramolecular cross-linkers. Property comparison on the basis of similar system would be more pronounced and convincing. To this end, we designed and prepared a new control sample (control-2) in which a complex containing a diplatinum acceptor coordinated with two bis(pyridinium) ligands (namely, a fragment of the molecular necklace cross-linker) could connect two DB24C8 moieties of the polymer to form [2]pseudorotaxane cross-linking points (newly added Scheme S4). Thus, the control-2 can be regarded as a general supramolecular network. The stress–strain curves exhibited that the mechanical properties of MIN-2 were superior to those of control-2 in all aspects including stiffness, strength, stretchability, and toughness (Fig. 4e). The better mechanical performance of MIN-2 benefits from the peculiar architectural and dynamic features of the molecular necklace: the multivalent interactions and reinforcing effect of rigid metallacycle skeleton endow the MINs with high stiffness, and the force-induced dynamicity of the interlocked structure guarantees the stretchability as well as the toughness of the MINs. And all these three features are responsible for the notable strength of the MINs. The detailed analyses about the relationships between structure of molecular necklace cross-linker and the mechanical properties of MINs have been discussed in the main text on pages 7 and 8. In addition, other measurements such as frequency sweep and the AFM modulus distribution profiles also supported the fact that MIN-2 had higher mechanical strength (Fig. 4c and newly added Supplementary Fig.36).

The stress relaxation experiments at different temperatures showed that the network of control-2 was destroyed at the temperature of 120 °C, but the MIN-2 kept intact at this temperature (newly added Fig. 5b and Supplementary Fig. 39). The better stability of MIN-2 may benefit from the mechanically interlocked structure. As we analyzed in the main text on page 9, both the literatures (Loeb, S. J. *et al. Org. Biomol. Chem.* **2006**, *4*, 667–680; Craig, S. L. *et al. Angew. Chem. Int. Ed.* **2005**, *44*, 2746–2748) and the stress relaxation experiments supported that the coordinate bond was much stronger than that of host–guest recognition. Therefore, although host–guest recognition in molecular necklace dissociated at a high temperature, the DB24C8 wheel was still able to be locked by the intact metallacycle, thus maintaining an effective network. Commonly, materials with supramolecular cross-links are noted for their dynamic properties, whereas they also suffer from relatively poor stability. Our molecular necklace cross-linked MINs could maintain usual dynamicity but improve the stability of network at the same time, representing an advanced dynamic cross-linker.

(2) From the perspective of microscopic working mechanism, the delicate dynamicity of molecular necklace cross-linker could realize a synergy of two kinds of supramolecular interactions. In addition to the experimental results of tensile and rheological tests, we further performed theoretical calculation (newly added Fig. 5d and Supplementary Fig. 43) to reveal the force-induced dynamicity of molecular necklace cross-link. Upon applied force, molecular necklace successively undergoes dissociation of host–guest recognition, sliding of DB24C8 wheel on the axle, bond deformation after reaching stopper, and dissociation of metal-coordination. The synergy of the two supramolecular interactions is that: the dissociations of host–guest recognition and coordinate bond occur in sequence, which are advantageous to make full use of the dual supramolecular interactions to enhance the mechanical properties based on the peculiar

topological structure. As we analyzed in the main text on page 10, although a large amount of energy is dissipated by the dissociation of host–guest recognition in molecular necklace, the cross-linking density of the network could keep intact because the DB24C8 wheel is confined in the metallacycle, which ensures a higher strength of the MINs. These features are difficult to be achieved by general dynamic cross-linked polymers even if there are two kinds of supramolecular interactions. The in-depth analyses have been described in the main text in the section of “force induced dynamicity of molecular necklace in MINs” on pages 8–11.

(3) Molecular necklace cross-linker has abundant dynamic properties due to dual supramolecular interactions, and more importantly, it could realize complicated and ingenious structural change upon stimuli. When evaluate the stimuli-responsiveness of a system, macroscopic property change of materials is admittedly crucial, but the microscopic structure change is also important from the point of chemistry. The MIN-2 treated with K^+ could de-thread the multiple DB24C8 moieties from the metallacycle but maintain the complete structure of the metallacycle. Though lots of stimuli-responsive polymers have been reported in literatures (Rowan S. J. *et al. Nat. Mater.* **2011**, *10*, 14–27; Huang, F. *et al. Chem. Soc. Rev.* **2012**, *41*, 6042–6065; Wang, S. *et al. Chem. Soc. Rev.* **2019**, *48*, 3537–3549), the cases which are able to achieve such a complicated structure change upon external stimuli are still rare.

In summary, our research intends to construct a fully new topological network and exploit the applicability of molecular necklace in cross-linking polymers based on its peculiar structural and dynamic features. Therefore, we paid more attention to three aspects: proving the formation of MINs, investigating their mechanical performances, and establishing the relationships between structures and mechanical properties. As we mentioned above, molecular necklace cross-linker possesses its unique advantages in many aspects compared with general dynamic cross-linking, and thus its application as a cross-linker is worth to be investigated. We believe that encouraged by the discovery of our research, materials cross-linked by molecular necklace with more novel and/or excellent properties would be developed in the future.

2. Comparing the mechanical strength in Fig. 6, it is clear that not necessarily only the crosslinking points by molecular necklaces affect the mechanical strength of the elastomer. It is important to clarify how efficiently the molecular necklaces are formed and to what extent they affect the mechanical properties.
 - We thank the reviewer for the kind reminder. The existence of molecular necklace cross-links in MINs had been proved both in solution and in the solid state as shown in Fig. 3. Subsequent property investigations based on tensile, frequency sweep, and stress relaxation experiments all supported that the molecular necklace cross-links are well formed in the MINs. Furthermore, it has been proved that the formation of the dynamic molecular necklace through self-assembly was highly efficient, especially with the presence of excessive DB24C8 units (Stang, P. J. *et al. J. Am. Chem. Soc.* **2014**, *136*, 5908–5911). In our work, the molar ratio of the DB24C8 moiety and bis(pyridinium) ligand was up to 4/1 for MIN-2, which was high enough to form molecular necklace

crosslinks according to the literature. Therefore, these evidences indicate that the molecular necklace cross-links could form efficiently in the MINs.

Before disclosing the effect degree of molecular necklace cross-links on the mechanical properties, the influence factors for the mechanical properties of the MINs should be summarized firstly. It can be revealed by comparing the stress–strain curves of pure polymer, two controls, and MIN-2. Here, we adopted Young’s modulus and breaking stress as indicators to analyze the influence factors. As shown in Fig. IIa, the pure linear polymer (Fig. Ia) was soft with extremely low modulus and strength (Fig. IIb and c). When it mixed with the bis(pyridinium) ligand and the diiodo-precursor of diplatinum acceptor to produce control-1 (Fig. Ib), the modulus and strength still enhanced obviously. This result indicated that the addition of rigid components was beneficial to enhance the stiffness and strength of the polymeric system even without forming cross-links. And this intrinsic influence of the rigid components on the mechanical properties of MINs should also be considered in the following analysis. The constitution of control-2 is highly similar to that of control-1 except that the diplatinum acceptor could coordinate with two bis(pyridinium) ligands to crosslink the polymer (Fig. Ic), whereas the modulus and breaking stress of control-2 were obviously higher than those of control-1, manifesting the role of cross-linking in determining the mechanical properties of polymer. Further upgrading the cross-linker to molecular necklace (Fig. Id), much superior mechanical properties were observed for MIN-2 compared with those of control-2. For the comparison of Young’s modulus, its evaluation was based on the linear region without damaging of molecular necklace, thus the enhanced stiffness of MIN-2 might be resulted from the multivalent cross-linking and the reinforcing effect of metallacycle skeleton. It is reasonable that these two features also responsible for the higher strength of MIN-2. Besides, the force-induced dynamicity of the interlocked structure also contributed to the strength because higher energy was needed to break the cross-linking in MINs compared with that in control-2 as analyzed in the main text on page 10. Therefore, the above comparisons revealed that: for MINs, the mechanical properties were not only decided by cross-linking in a general sense but also affected by many other features of molecular necklace including the reinforcing effect of metallacycle skeleton, the multivalent interactions, and the force-induced dynamicity of mechanically interlocked structure.

The effect of the molecular necklace on the mechanical properties of MIN-2 could be considered as a synergy or a sum of the above-mentioned features. After knowing all the influence factors, the extent of mechanical properties contributed by the whole molecular necklace cross-link could be assessed by the comparison between MIN-2 and control-1. Because cross-linking as well as all the other features of molecular necklace are absent in the control-1, but the structural elements of control-1 are similar to MIN-2, and thus the intrinsic influence originated from the discrete rigid components of bis(pyridinium) ligand and diplatinum acceptor could be excluded. As such, the enhanced mechanical properties of MIN-2 compared to control-1 could be ascribed to the contribution from the molecular necklace cross-links. Therefore, the extent of mechanical properties improved by the molecular necklace could be roughly evaluated by the difference of mechanical properties between MIN-2 and control-1.

Based on the analyses above, the results of tensile tests in Fig. 6 for stimuli-responsive experiments could be well interpreted. The sample treated with KPF_6 led to the separation of the metallacycle and the polymer, and the metallacycle was intact. In this case, the reinforcing effect of the rigid metallacycle still contributed to the strength of the system. Besides, there might be some steric interactions between the bulky metallacycle and polymer chain to affect the mechanical properties. For example, it is possible that multiple chains are locked by a metallacycle. But for the sample treated with TBABr , the Br^- could not only break the metal-coordination but also dissociate the host-guest recognition owing to the exchange of counter anions from PF_6^- to Br^- which doesn't support host-guest recognition. As a result, the MIN-2 decomposed into three independent components: a pure polymer, a bis(pyridinium) ligand, and a diplatinum acceptor. Due to the lack of effective interactions among the three components, tensile behaviors of the mixture are thus similar to the pure polymer rather than the control-1 in which the DB24C8 wheels on the polymer backbone can recognize the bis(pyridinium) ligands. Notably, the salts of KPF_6 and TBABr were reserved in corresponding specimens after stimuli-responsive experiments, which might also have an influence on the mechanical strength of the specimens. For example, the DB24C8 moiety on the polymer could bind a K^+ to form a more rigid unit, thus strengthening the specimen in a certain degree. In a word, the distinct changes of the mechanical properties for MIN specimens upon stimuli clearly exhibited the abundant dynamic properties of the molecular necklace cross-linked materials.

Fig. I Constitutions and chemical structures of pure polymer (a), control-1 (b), control-2 (c), and MIN-2 (d).

Fig. II a Stress-strain curves of pure polymer, controls, and MIN-2 recorded at room temperature with a deformation rate of 100 mm/min. Young's modulus (b) and maximum stress (c) of the samples calculated from their stress-strain curves.

- Evaluation of mechanical strength of organogel swollen with THF helps to understand the correlation between structure and mechanical properties.

- We thank the reviewer for the kind reminder of this point. As we know, the polymeric networks in swollen state are able to weaken the inter-chain interactions such as frictional force, which is advantageous to highlight the role of molecular necklace in mechanical behaviors. For this purpose, we performed a series of tensile and rheological measurements for MIN-2 swollen with THF (swelling ratio ~1.7 g/g), and compared the results with those of the bulk materials to further understand and verify the relationships between molecular necklace cross-linker and mechanical properties of MINs.

In the frequency sweep tests (Fig. III a), the storage modulus of the swollen sample was always higher than the loss modulus over the entire frequency range, which suggested that the network structure was well preserved in the swollen MIN-2, verifying the role of molecular necklace in cross-linking. Meanwhile, compared with the bulk MIN-2, the shear moduli of the swollen sample were much lower, which could be ascribed to the decrease in number densities of effective chains due to swelling.

Stress–strain curves (Fig. III b and c) provided a more sharp contrast between the bulk and swollen MIN-2 specimens. Similar to the frequency sweep results, the Young's modulus of the swollen sample (3.22 MPa) was much lower than that of bulk material (47.3 MPa), and the breaking stress also had the same tendency. These phenomena were common for the swollen samples (Morita, H. *et al. J. Phys. Soc. Jpn.* **2009**, 78, 041008). Besides, the breaking strain of the swollen sample (~500%) was also slightly lower than that of bulk MIN-2 (~600%) because the polymer chains were prestretched by swelling in THF, which was also observed in the case of rubber with covalent cross-links (Morita, H. *et al. J. Phys. Soc. Jpn.* **2009**, 78, 041008). By comparison, the breaking strain of the swollen sample in our work remained a relatively high level. The different swelling ratios should be a possible reason, but the dynamicity of molecular necklace cross-link could also contribute much to the notable stretchability.

To reveal the specific role of molecular necklace in swollen MIN-2, cyclic tensile tests were also conducted (Fig. III d and e). Similar to bulk MIN-2, the initial circle showed a large hysteresis area, indicative of a pronounced energy dissipation. Furthermore, the second circle without rest exhibited a significant residual strain as well as a markedly reduced hysteresis loop, which could be interpreted by the delaying recovery of the supramolecular interactions. As mentioned above, the swollen sample could reduce the effect of interactions between polymer chains. Therefore, these results are more favorable to prove that the molecular necklace could respond to applied force through its structural change to dissipate energy and thus toughen the MINs.

The response speed of molecular necklace cross-link upon force stimulus could be reflected by the strain sweep tests (Fig. III f). It is well-known that the linear viscoelastic region of the strain sweep curve could be used to evaluate the critical strain where the network structure starts to destroy. The destroyed structure in our work should be related to the damage of the molecular necklace cross-link. Here, we selected the tolerance range of 10% deviation for G' around the plateau value as the standard to assess the strain, and the values for swollen and bulk MIN-2 samples were measured to be 1.0% and 2.5%, respectively. The lower strain value for the swollen sample indicated a faster response of the molecular necklace cross-link. Because the chain in the swollen sample was prestretched by the solvent and thus the force could concentrate on the molecular necklace

more quickly. Based on these results, we also speculate that: for the bulk MIN-2, the initial 1.5% (2.5% – 1.0%) shear strain is mainly attributed to the stretching of polymer chain, and after that, the force starts to concentrate on the molecular necklace cross-link.

In summary, through the property investigation of the swollen sample, the effective cross-linking role of molecular necklace was further proved. Moreover, benefiting from reducing the effect of inter-chain interactions, the dynamicity of the molecular necklace upon stress and its influence on the stretchability and toughness of the MINs were revealed more clearly. In particular, more detailed information about the shear strain responsiveness of the molecular necklace cross-link in bulk MIN-2 could be assessed based on the strain sweep comparison between the bulk and swollen MIN-2 samples.

Fig. III (newly added Figure S38) **a** Frequency sweep of bulk MIN-2 (40 °C) and swollen MIN-2 (25 °C). **b** Stress–strain curves of bulk and swollen MIN-2 samples recorded at room temperature with a deformation rate of 100 mm/min. **c** Young's moduli and maximum stress of bulk and swollen MIN-2 samples calculated from their stress–strain curves. **d** Cyclic tensile test curves of swollen MIN-2 loaded at a strain of 300% recorded at room temperature with a deformation rate of 100 mm/min. **e** Hysteresis area for the two consecutive cycles of the tensile tests for swollen MIN-2. **f** Strain sweep curves of bulk MIN-2 (40 °C) and swollen MIN-2 (25 °C) at a constant angular frequency of 1.0 rad/s.

4. The correlation between sample name and curve in DSC in Fig. S28 is not reasonable.
- We appreciate the kind reminder. The figure has been modified and updated accordingly.

For Reviewer 2:

1. This manuscript reports a mechanically interlocked networks (MIN) by crosslinking a linear polymer with crown ether side chains and metallacycle via coordination-driven self-assembly. Dynamic crosslinks are formed by host-guest interaction and dynamic dissociation and reassociation of macrocyclic coordination bonds. The mechanically

interlocked network shows various peculiar mechanical responses based on the formation, sliding motion, and dissociation of dynamic crosslinks. Since the manuscript is well written with the unique mechanically interlocked structure, I think that it might be published in Nature Communication.

- We thank this reviewer for the positive comments.
2. However, it seems like there are some unclear explanations about the mechanical data. The authors said that no significant cross-links exist in the control sample. Certainly, the stress relaxation in Fig. 4d indicates that but the viscoelastic profile of the control sample in Fig. 4c shows a frequency dispersion like physical crosslinks, which is different from that of the polymer melt, in the frequency range measured. The same frequency dependence is observed in MIN-2 as well. The authors should explain the origin of the frequency dispersion.
- We appreciate the kind reminder. It is true that there is a pronounced frequency dispersion for control-1 in the measured frequency range. To find out the reason, we firstly confirmed the fact that coordination between the bis(pyridinium) ligand and the diiodo-precursor of diplatinum acceptor was not possible. For this purpose, we mixed the two components in solution and measured corresponding NMR spectra after 2 days. As shown in Fig. IV, all peaks belonging to the two components were well reserved in the spectrum of the mixture, and meanwhile, no peaks shift or new signals were observed. These results supported the conclusion that the two components in control-1 were unable to form cross-linking point through metal-coordination. A more plausible explanation for the frequency dispersion would seem to be the slight entanglement of polymer chains. In our previous work, we found that slight entanglement occurred for the pure polynorbornene derivative with molecular weight of 75 kDa (Yan, X. *et al. Angew. Chem. Int. Ed.* **2020**, *59*, 12139–12146). Furthermore, similar to control-1, obvious frequency dispersion was also observed in its frequency sweep experiment, and it also relaxed the stress rapidly during the stress relaxation measurement. As for the DB24C8-functionalized polynorbornene derivative in our work, its molecular weight is about 77 kDa. Therefore, it is reasonable to speculate that slight entanglement would be present in the control-1 sample. According to the reminder of reviewer, we also added corresponding discussion in the main text on page 7 to describe the mechanical data more clear and reasonable. We thank the reviewer again for this kind reminder.

Fig. IV (newly added Figure S29) Partial ^1H NMR spectra (acetone- d_6 , room temperature, 400 MHz) of the diiodo-precursor (a), the mixture of diiodo-precursor and bis(pyridinium) ligand (b), and the bis(pyridinium) ligand (c).

3. In the molecular mechanism of dynamic crosslinks in MIN in Fig. 5e, I feel that there are no clear evidence supporting the second process: the dissociation of host-guest recognition and sliding. From the mechanical data, it seems to be difficult to discriminate between the second and third processes. Is it possible to explain which data clearly shows the existence of the second process?
- We thank the reviewer for this reminder. As the reviewer pointed out, the microscopic dissociation of the host-guest recognition and subsequent sliding are difficult to be unambiguously exhibited by the macroscopic mechanical properties. And corresponding conclusions are our reasonable deduction based on the experimental data as well as the relevant literatures (De Bo, G. *J. Am. Chem. Soc.* **2019**, *141*, 15879–15883; Yan, X. *et. al. J. Am. Chem. Soc.* **2022**, DOI: 10.1021/jacs.1c10427). Moreover, during the revision, we further performed theoretical simulation to verify the speculation and accurately describe the dynamic process of molecular necklace upon stress. The detailed evidences are summarized as follows:

According to the literatures, the association constant of the host-guest recognition between DB24C8 host and bis(pyridinium) guest was reported to be 930 M^{-1} (Loeb, S. J. *et. al. Org. Biomol. Chem.* **2006**, *4*, 667–680), which was much lower than that of the metal-coordination between platinum and pyridine ligand (4000 M^{-1} in the literature of Craig, S. L. *et. al. Angew. Chem. Int. Ed.* **2005**, *44*, 2746–2748). Therefore, compared with the host-guest recognition, the strength of the metal-coordination is much higher. This conclusion was consistent with our experimental results. In the stress relaxation experiments at different temperatures, the control-2 specimen relaxed the stress rapidly and completely at $120\text{ }^\circ\text{C}$ (newly added Supplementary Fig. 39), whereas notable stress was kept for MIN-2 at the same temperature (Fig. 5b). The complete stress relaxation in these two samples could be related to the cleavage of the physical cross-links and

consequent destroy of the network structures. For control-2, the result meant that the complete dissociation of host–guest recognition occurred at 120 °C. In MIN-2, though the high temperature of 120 °C also led to the dissociation of host–guest recognition, the dissociated DB24C8 was still confined in the metallacycle, thus maintaining an effective network.

The above analyses have indicated that the strength of metal-coordination is much higher than that of host–guest recognition. Hence, it is reasonable to speculate that upon stress, the dissociation of host–guest recognition would take place firstly, and after that, the DB24C8 wheel would slide on the axle to stop at a site where the applied force could effectively transfer to the whole metallacycle to further cleave the metal-coordination. This speculation was supported by the recovery experiments based on cyclic tensile tests (Fig. 5c). The results showed that the mechanical properties of the sample after being stretched cannot get recovered completely. As we analyzed in the main text on page 9, the unrecovered mechanical properties might be related to the incomplete recovery of host–guest recognition. Because certain percentage of DB24C8 moiety was pulled away from the metallacycle upon force, and after unloading, the coordinate bonds were relatively easy to restore but rethreading of the dissociated DB24C8 moieties onto the metallacycle were much difficult in the solid state. Therefore, the cyclic tensile results are in a good agreement with the deduction that dissociation and sliding occurred during the tensile experiments.

More importantly, the analyses above were further confirmed by the calculation using the constrained geometries simulate external force (CoGEF) method. As shown in the newly added Fig 5d, the CoGEF elongation energy curve could be divided into three parts which are corresponded to the gradual dissociation of host–guest recognition, the sliding of the DB24C8 wheel on the axle, and a sharp increase in energy after the wheel reachesthe stopper followed by the dissociation of coordinate bond, respectively. Corresponding structures at different stages were shown in newly added Supplementary Fig. 43. Thus, the occurrence of dissociation of host–guest recognition and sliding of DB24C8 on the axle were clearly displayed by the CoGEF simulation. In addition, more details about these processes could be revealed by the method. For example, the energy to dissociate the host–guest recognition (237 kcal/mol) was about one-third of that for the metal-coordination (670 kcal/mol), and the travel distance of the DB24C8 wheel on the axle after dissociation was about 4~5 Å. The in-depth analyses of the CoGEF calculation and the dynamicity of molecular necklace were discussed in the main text on page 10.

REVIEWER COMMENTS

Reviewer #1 (Remarks to the Author):

My concerns were addressed with ample explanation and experimental data. A network of molecular necklace cross-links will be expected to create a new trend in materials design.

Reviewer #2 (Remarks to the Author):

The authors replied to my comments and questions properly, which satisfied me sufficiently. I think that it can be published in Nature Communications now.